# Comparing a computational model of visual problem solving with human vision on a difficult vision task

**Tarun Khajuria**⬡*, **Kadi Tulver, Jaan Aru**

Institute of Computer Science, University of Tartu, Tartu, Estonia

* tarun.khajuria@ut.ee

## Abstract

Human vision is not merely a passive process of interpreting sensory input but can also function as a problem-solving process incorporating generative mechanisms to interpret ambiguous or noisy data. This synergy between the generative and discriminative components, often described as analysis-by-synthesis, enables robust perception and rapid adaptation to out-of-distribution inputs. In this work, we investigate a computational implementation of the analysis-by-synthesis paradigm using genetic search in a generative model, applied to a visual problem-solving task inspired by star constellations. The search is guided by low-level cues based on the structural fitness of candidate solutions compared to the test images. This dataset serves as a testbed for exploring how inferred signals can guide the synthesis of suitable solutions in ambiguous conditions, framing visual inference as an instance of complex problem solving. Drawing on insights from human experiments, we develop a generative search algorithm and compare its performance to humans, examining factors such as accuracy, reaction time, and overlap in drawings. Our results shed light on possible mechanisms of human visual problem solving and highlight the potential of generative search models to emulate aspects of this process.

## Author summary

Human vision is not just about passively receiving information from the environment. Rather, it also involves actively making sense of what we see. When faced with unclear or incomplete visual input, our brains use prior knowledge to fill in gaps and create the most likely interpretation. This ability helps us recognize objects and patterns even in difficult conditions. In this study, we explore how this process can be replicated using computer models. Specifically, we test a method that generates possible interpretations of ambiguous visual data, inspired by the way people recognize star constellations. By comparing the model's performance with human participants, we examine how well it mirrors human perception.

**Data availability statement:** The code and data will be available at: https://github.com/tarunkhajuria42/GenSearch.

**Funding:** TK, KT and JA are supported by the Estonian Research Council grant PSG728. JA is also supported by Estonian Research Council grants Tem-TA 120 and the Estonian Centre of Excellence in Artificial Intelligence (EXAI) funded by the Estonian Ministry of Education and Research. KT is also supported by Estonian Research Council grant PUTJD1252. The funders had no role in study design, data collection and analysis, decision to publish, or preparation of the manuscript.

**Competing interests:** The authors have declared that no competing interests exist.

We analyze factors such as accuracy, response time, and similarities in the interpretations produced. Our findings offer insights into how people make sense of uncertain visual information and suggest ways in which computer models can be designed to mimic this ability. These results can contribute to our understanding of human perception as well as help advance artificial vision systems beyond simple pattern recognition.

## Introduction

Human visual processing is adapted to operate in dynamic and complex environments. This means that human vision has to solve a context-based scene analysis problem, where humans iteratively obtain and process information from the scene to refine their understanding of the scene, which becomes their new context [1,2]. In this way, vision becomes more like problem-solving, where the solution is not known in advance and one has to iteratively probe several alternatives. One way to understand the nuances of this process is to develop machine vision systems that can replicate this dynamic, iterative, context-driven hypothesis exploration and refinement algorithm on difficult tasks.

Many computational frameworks have been proposed to explain dynamic aspects of vision. For example, Yuille and Kersten [3] highlighted the need to study vision using stimuli with the dynamics and complexity of the real 3D world. They also promoted analysis-by-synthesis, i.e., using a top-down signal to guide bottom-up search under Bayesian inference as a prime candidate to explain these real-world complexities. In processing highly degraded images, analysis-by-synthesis has been a primary candidate algorithm used to explain human performance. An analysis-by-synthesis algorithm was found to be a better match to human performance in identifying highly degraded Mooney images compared to conventional feed-forward deep learning algorithms [4]. Yildirim and colleagues [5] discussed the problems with many analysis-by-synthesis solutions, including their inability to scale to many real-world problems and being too slow to explain human vision. Further, they proposed a solution to this problem with a fast analysis-by-synthesis algorithm to work on the difficult problem of 3D face perception by using a single photo that can be obtained from various camera poses. In images of natural scenes, [6] introduced parsing the scene in multiple iterations to model the components and their relations and create a scene graph. Ullman and colleagues [7] proposed a solution to scene analysis using a goal-guided algorithm that combines top-down and bottom-up processes for understanding a scene. This algorithm uses a convolutional neural network (CNN) to extract bottom-up features and then produce a top-down command token that guides the re-extraction of other bottom-up features in the next step. The system iterates between using the bottom-up and top-down network to focus step by step on relevant parts of the image and creates a scene graph as required by the goal of the task. Further, [8] used similar insights from analysis-by-synthesis to inspire changes to attention mechanisms in modern DL architectures, which improved performance across tasks incorporating this context-driven top-down attention.

In light of this progress in the application of the analysis-by-synthesis framework, we specifically wanted to understand the process humans follow when solving a complex vision problem. To study this question, we use the constellations image task introduced in [9] where objects are hidden similarly to constellations of stars. To create these images, object outlines are used to form a dotted shape and extra noise dots are added as distractors (Fig 2a). The task is to identify the object in the scene by tracing the dots that form the object's contours. This approach allows us to use static images for a dynamic and complex image processing task involving iterative steps that mimic problem-solving tasks in other domains. For instance, trying to identify the approaching person in a dark alley is akin to a problem-solving task, as it requires combining low-quality bottom-up cues with prior knowledge and iterating over many possibilities. The task also adheres to the principles of analysis-by-synthesis as it involves both a bottom-up objective of making sure that the drawn outlines pass through the dots and a top-down goal of generating possible objects under these constraints. The discrete task of following the dots makes it challenging to train models using gradient descent. However, it also allows progress to be measured clearly by simply counting the continuous lines passing through the dots. This approach helps to analyse the drawings during the solving process and provides a better understanding of the iterative process involved in solving these images.

We propose a generative search algorithm (hereafter referred to as GenSearch) guided by structural cues from the images to capture the interplay of bottom-up identification of regularities in the image followed by a top-down search to find the complete drawing. We use genetic search for possible images in an image generator's (GAN's [10]) latent space that fits the structure of dots in the images. We perform experiments to analyse the performance of humans and the GenSearch algorithm on two sets of constellation images generated from MNIST [11] and Fashion MNIST [12]. The solving process was captured by allowing humans to draw their guesses while in the process of solving the images and for the algorithm to store the intermediate solutions. Our analysis shows an alignment of the GenSearch algorithm to humans regarding performance, solving processes, drawing alignment, and the confusion between similar shapes. Understanding the model's inner workings gives us insights into the possible processes humans might follow during complex visual problem-solving.

## Results

In this section, we first describe the generative search algorithm (GenSearch) that we developed for solving the constellation images. Next, we discuss the differences in overall classification performance between the GenSearch algorithm and human participants. As both humans and GenSearch generated drawings while solving the images, we analyse their drawings and compare their solving processes. Lastly, we draw comparisons with the baseline performances of several Deep Learning (DL) models in their attempts to solve the constellation images.

### Proposed Generative Search (GenSearch) algorithm

This section discusses the details of our proposed GenSearch Algorithm 1. In designing our search algorithm, we operationalised the solution to the constellation images as the process of drawing the outline of the object, followed by an evaluation of its fit to the actual shape (see Fig 1). To implement this, we propose an evolutionary search in the generator's latent space to generate likely shapes, with the search guided by the fitness (number of dots the contour passes through) evaluated on the constellation image. The generator is used to create candidate images which are then converted into outline contours to check if they fit with the dot structure in the test image. A CNN classifier finally classifies the generated images (not the outlines or edges) into a relevant image category. The details of the evolutionary search algorithm are described in the steps below.

The search is implemented as a continuous space evolutionary search with the following steps:

1. The initial population of solutions is randomly sampled from an image generator's latent space. We then sample a set of vectors in the latent space of the GAN and then use them to generate the candidate images with the GAN.

**Algorithm 1 Generative Evolutionary Search Algorithm for solving constellations (GenSearch).**

1: *ConstellationImage* = Input constellation image to solve
2: **Initialize Population:**
3: Sample Population $\{z_1, z_2, \ldots, z_N\}$ from the latent space of Generator where $N = 1000$.
4: **for each generation** $g$ = 1 to *MaxGenerations* = 30 **do**
5: **a. Evaluate Fitness for Each Solution:**
6: **for each** latent vector $z_i \in Population$ **do**
7: Generate image $Image_i = Generator(z_i)$
8: Convert $Image_i$ to greyscale
9: Detect edges in $Image_i$ using edge detection (e.g., Canny) $\rightarrow$ *Contours*
10: **for each** dot $D \in ConstellationImage$ **do**
11: **if** $D$ has distance to edge in $Image_i$ < *EdgeThreshold* = 3 **then**
12: Add contribution to fitness score
13: **end if**
14: **end for**
15: Store $Fitness_i$ for $Image_i$
16: **end for**
17: **b. Select Top Solutions:**
18: Rank all solutions by $Fitness_i$
19: Select top $K$ = 200 solutions: $TopSolutions = \{z'_1, z'_2, \ldots, z'_K\}$
20: **c. Generate New Population:**
21: Create offspring by applying:
22: **Point Mutation:** Randomly alter features of $z'_i$
23: **Crossover:** Combine features of two parent vectors $z'_i$ and $z'_j$
24: Form new Population with size $N$ from offspring
25: **d. Track Best Solution:**
26: $BestSolutionGeneration[g] = z_{best}$ where $z_{best}$ has the highest fitness in this generation
27: **Check Convergence:**
28: Compute change in fitness score of $BestSolutionGeneration[g]$ over generations
29: **if** fitness over 5 generations not updated **then**
30: **Terminate**
31: **end if**
32: **end for**
33: **Output the Final Best Solution:**
34: $Image_{best} = Generator(z_{best})$
35: **return** $Object = Classifier(Image_{best})$

2. The fitness for these solutions is calculated by comparing the generated image's fit to the dots on the constellation image.

3. The calculation of an object's fit to the dots constitutes fitting the edges of the object contour to the dots. Hence, we first run an edge detection on the sample's greyscale image and then use the obtained contours to check if they pass through a dot. We allow a tolerance of 3 pixels for assessing connectivity. Only dots connected by contour edges less than 40 pixels apart contribute to the fitness score, preventing the model from favouring excessively long, unrealistic drawings.

4. Finally, after the top 200 images from each population are selected based on their fitness, we generate the new population by point mutation (change of genes randomly) and crossover of the features (combination of genes from two parents, i.e. two of the top 200 candidate vectors) of these vectors. The process is repeated for 30 generations.

5. At the end of each generation, the solution with top fitness is considered the solution of that iteration (generation). The convergence is calculated based on the change in the fitness of top solutions over the generations.

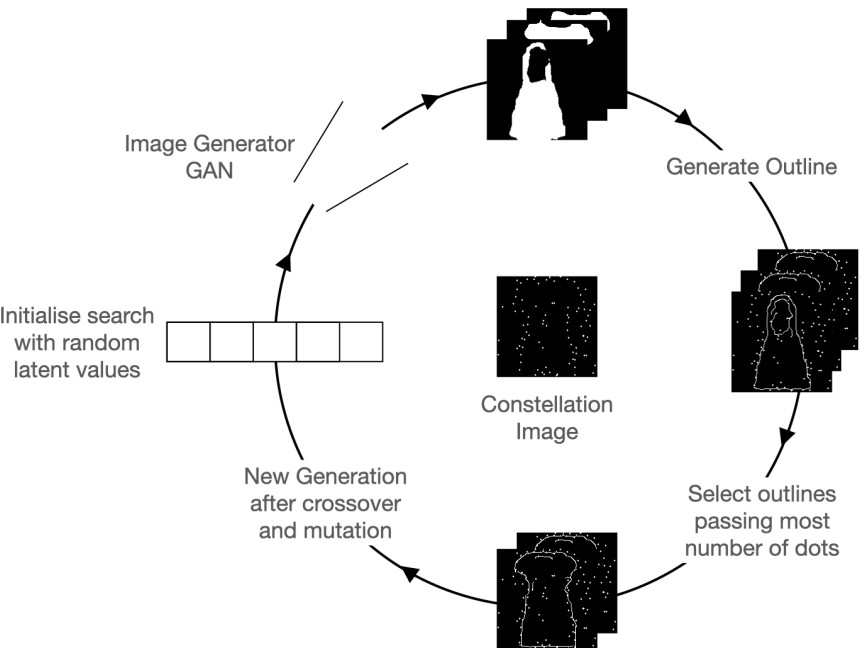

**Fig 1**. **GenSearch algorithm.** The constellations image is solved by generating candidate solutions with a GAN and refined using a genetic search conditioned on best fitting of the solution outlines to the dots on the constellation image.

## Comparison of object classification performance between GenSearch and humans

We analysed the performance of the proposed search algorithm on the constellation images from two datasets: 1) MNIST constellations and 2) Fashion MNIST constellations. These two datasets consist of constellation images generated from images of numbers and fashion items, respectively. Both sets have 10 categories each. In both cases, the train set consists of 60,000 images and the test set consists of 10,000 images. Out of the 10,000 images, we selected 38 images suitable for human experiments and compared them against the computational model.

As the primary task that can be evaluated using this dataset is to identify the object hidden in the constellations and draw its outline, we first compare the identification accuracy between GenSearch and humans on both datasets. In Fig 2b, we can see that the classification accuracy is comparable between humans and GenSearch across both datasets, with both humans and the model correctly identifying around 60% of the objects.

The search model's mistakes are also comparable to those of humans, as both tend to confuse objects with similar outlines, such as t-shirts, pullovers, and shirts or 4s and 9s, etc. See in Fig 2c the confusion matrix for the GenSearch and Humans. Pearson's correlation between the human and model's confusion matrices (excluding zeros) for Fashion MNIST is 0.79. In the case of MNIST, the model confuses between similar digits, and the correlation to the humans' confusion matrix is 0.70. Since 11 humans solved the same image for the Fashion MNIST dataset and 10 for the MNIST dataset, the total counts in the human confusion matrices are correspondingly higher by that factor.

## Comparison and analysis of the solution process of GenSearch and humans

In addition to a comparison of object classification performance, we were interested in how aspects of the solution process would compare between our search algorithm and human process. In the next section, we will discuss qualitative measures and descriptions related to the process of solving constellation images, including testing different hypotheses by drawing on the image and evaluating the fit of the drawings compared to the dots.

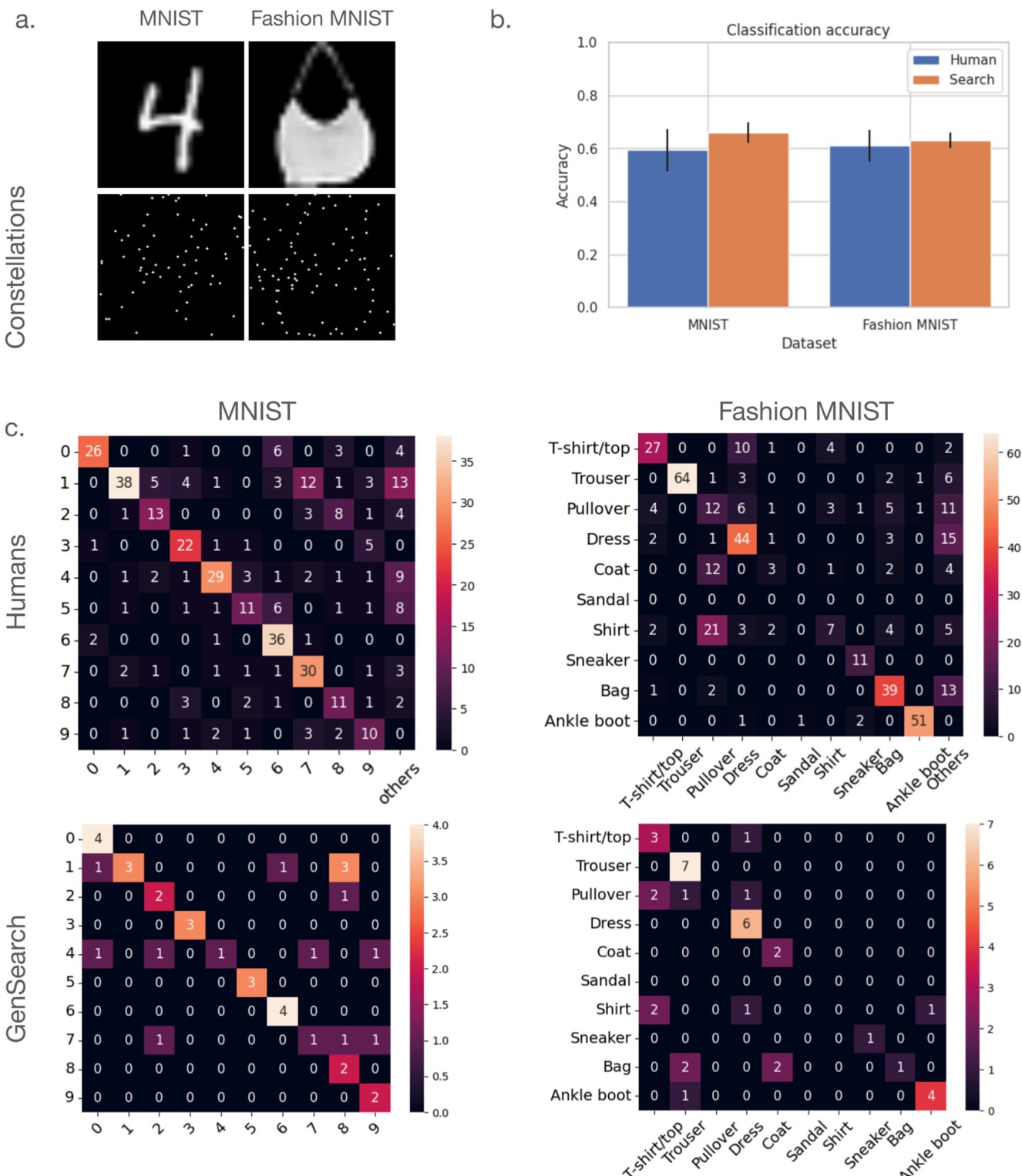

**Fig 2**. **Classifying objects in constellation images.** a) Sample images from the original Fashion MNIST and MNIST datasets along with their constellations version below them. b) The plot shows the comparable classification accuracy of Humans and GenSearch algorithm on a test set of 38 images on both datasets c). Confusion matrix for classification in 1) Fashion MNIST for Humans 2) Fashion MNIST for GenSearch. The correlation between

the two matrices is (Pearson's coefficient) 0.79. Confusion matrix for 3) MNIST for Humans and 4) MNIST for Search Algorithm with a correlation of (Pearson's coefficient) 0.70 between the matrices. We remove the entries when the corresponding elements are 0 in both matrices to calculate the correlation. Additionally, in many cases, humans don't respond or give a response that does not correspond to any of the given options. Such responses are aggregated in the column 'Others'.

**Overlap between the drawing solutions.** The solution process between GenSearch and humans matches in specific ways. We see an overlap between the drawings made by humans and the generative model. For images in which humans and the GenSearch algorithm wrongly identified the object, the drawings capture similar contours in many of these images, hinting at similar patterns in the bottom-up processing of dot patterns. We quantify the overlap between the human drawings and the GenSearch solution by calculating the intersection over union IOU(dots) in terms of the constellation points covered by the drawings. An average IOU is calculated over the whole dataset to quantify the overlap (see Fig 3). The average IOU for Fashion MNIST and MNIST datasets is 0.49 and 0.60, respectively. This shows a high average overlap between the images. Additionally, we quantified the overlap only between the mistakes made by humans and GenSearch by removing the ground-truth points from the IOU calculation, this metric is called IOU(mistakes). These numbers are also high, with 0.58 for Fashion MNIST and 0.43 for MNIST datasets.

**Similarity in types of mistakes.** We observe that humans and GenSearch make similar mistakes as they confuse many objects with similar overall shapes. In Fig 4, we see examples of such mistakes for both datasets by humans and GenSearch. The common type of mistake is where the particular edge or feature is missed or added by GenSearch or humans, changing the category of the solution. This is very common (in 7 out of all 13 misclassifications by GenSearch) in MNIST, as the digits in many forms differ from each other by a single edge. The example given in Fig 4 is where the top of the 4 is so close that both GenSearch and humans complete the circle to make a 9. In Fashion MNIST, this

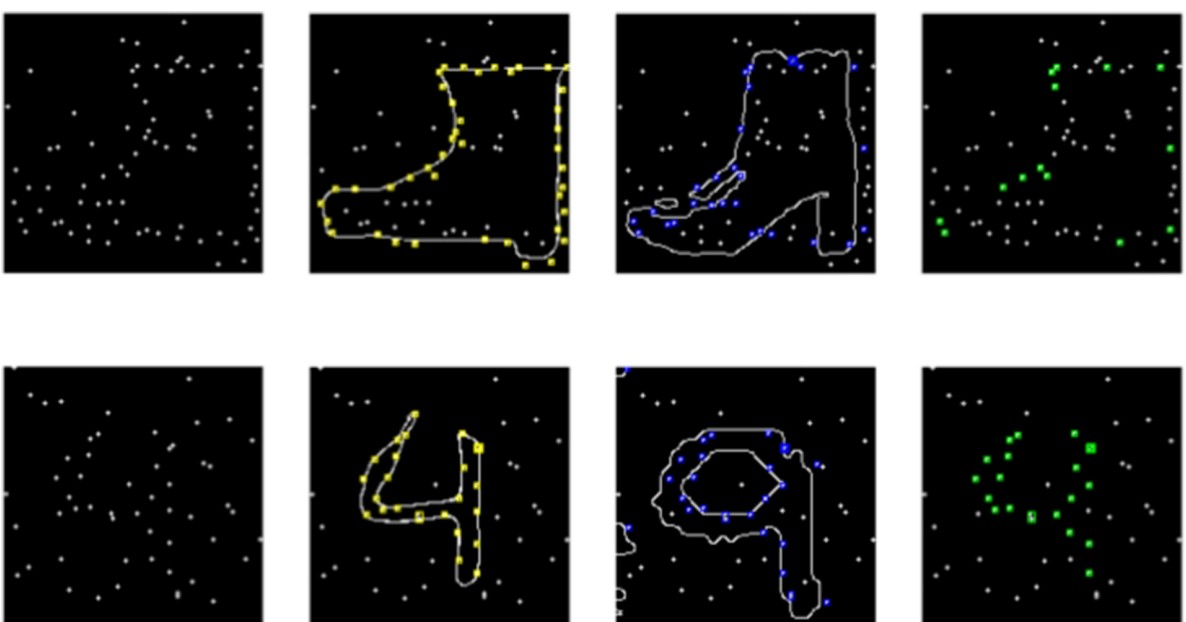

**Fig 3**. **Quantifying solution overlap using IOU (dots).** Examples from Fashion MNIST (top) and from MNIST (bottom), from left to right 1) Constellation image 2) dots covered by human drawing 3) dots covered by search algorithm drawing 4) common dots between search and human drawing. The intersection over union (dots) for this Fashion MNIST is 0.25, whereas IOU(dots) for MNIST is 0.8. The overall IOU(dots) for the whole dataset for human drawings to the corresponding search solution is 0.49 for Fashion MNIST and 0.6 for the MNIST dataset.

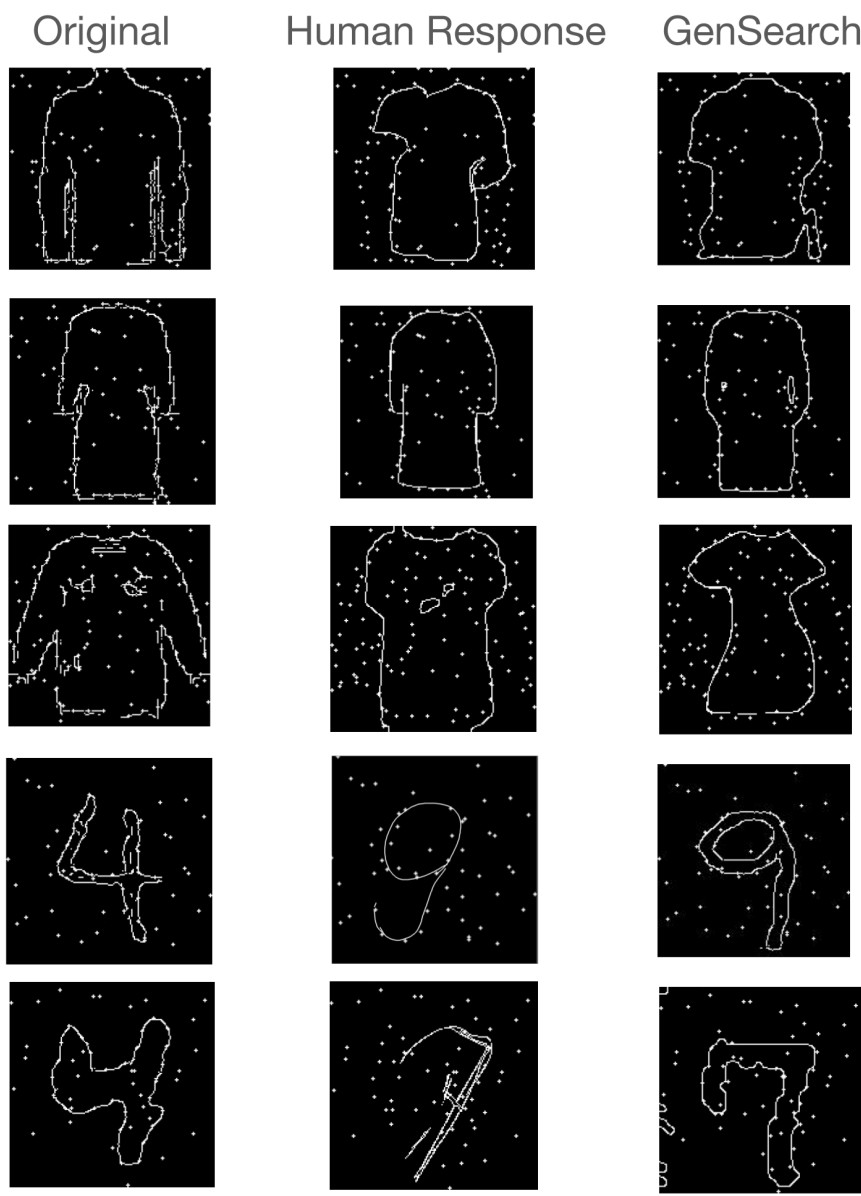

**Fig 4. Similarity of misclassified classes in humans and GenSearch.** The top 3 panels illustrate mistakes in Fashion MNIST constellations, where both GenSearch and humans confuse between pullover, t-shirt, dress and a coat. The lower two panels show how the number 4 is confused with a 9 and a 7. We also observe that beyond mistaking one category for another, the pattern of mistakes also involves following a particular shape incorrectly or missing a particular edge while having fair overlap on other parts of the shape, as shown in the example of the pullover.

phenomenon occurs many times (in 7 out of 14 total misclassifications by GenSearch) between some categories such as T-shirts, Pullover, Shirts and Dresses or between Sneaker and Ankle boots.

**Iterative refinement of solutions.** Beyond the overlap of solutions and the commonalities in mistakes, we observe that the search processes of both humans and the GenSearch algorithm look similar in many ways. For GenSearch, we observe the iterative refinement of the top solution over the generations. There are always phases during the solving process where the top solution remains relatively stable over generations while gradually adjusting its shape to maximise the number of dots it passes through. As shown in Fig 5, a shoe with a heel can change to a flat shoe - such alterations

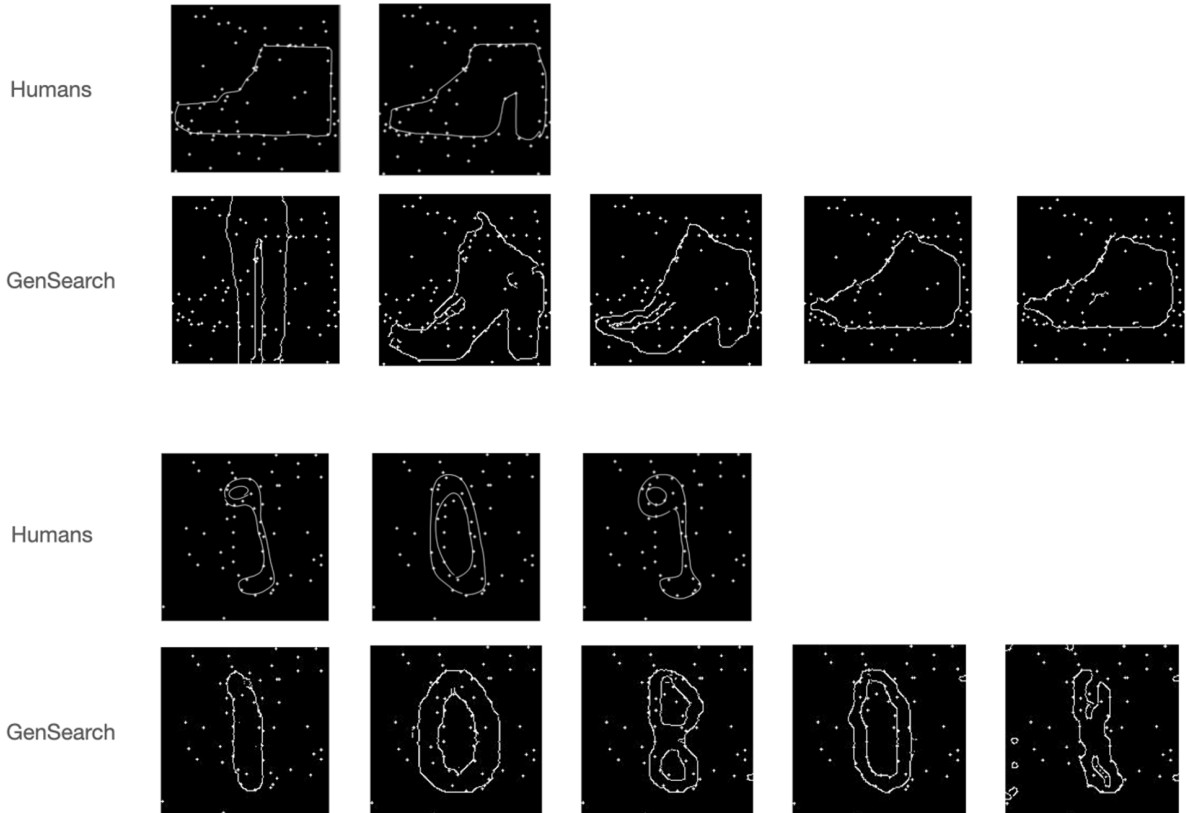

**Fig 5**. **Intermediate solutions.** Qualitatively, the process of solving for humans and GenSearch looks similar as both processes iteratively change their top solution in the search process. The figure depicts an instance where both humans and GenSearch explore similar candidate solutions. The correct solution for the top image is a shoe and for the bottom image is the number 1

preserve the most aligned features while trying out different variations. This can be contrasted with cases where the top solution undergoes a sudden shift, resulting in a completely different category emerging as the top solution, which is then further refined. In most cases, the top solutions are refined with minor adjustments in later generations. Similarly, humans displayed both iterative refinement of hypotheses as well as instances of a sudden switch to a solution (also reported in [9]). An overall comparison of human solving time to GenSearch's convergence is given in S2 Text.

**Possibility of maintaining multiple hypotheses resulting in the sudden emergence of a correct solution.** In the previous section, we discussed instances of a sudden change in the top solution. However, this shift in the top solution is not necessarily sudden from the perspective of the search process, as this candidate solution is maintained and refined by the GenSearch algorithm in its previous generations. We note here that in each generation, the algorithm chooses not just the top solution but rather the top 200 solutions that contribute to the creation of candidates for the next generation. This process allows for a variety of good solutions to be preserved and refined in the background. We illustrate this process in Fig 6 by showing five solutions from the top 200 solution pool. Note how variants of the top solution, i.e. number 2, already exist in the first generation but it only emerges as a top solution by generation 25. Across generations, the solution gets refined and its shape adjusted, and as many of its variants fit the dot pattern quite well, they start dominating the solution pool by generation 15. As a result of this continuous refinement in the background, it emerges as the top solution in the later stages. We also note here that in the later stages, i.e. in generation 25 and 30, even if the candidate space is dominated by variations of 2, other solutions like 3 and 7 are still represented, allowing for further changes in the top solution in

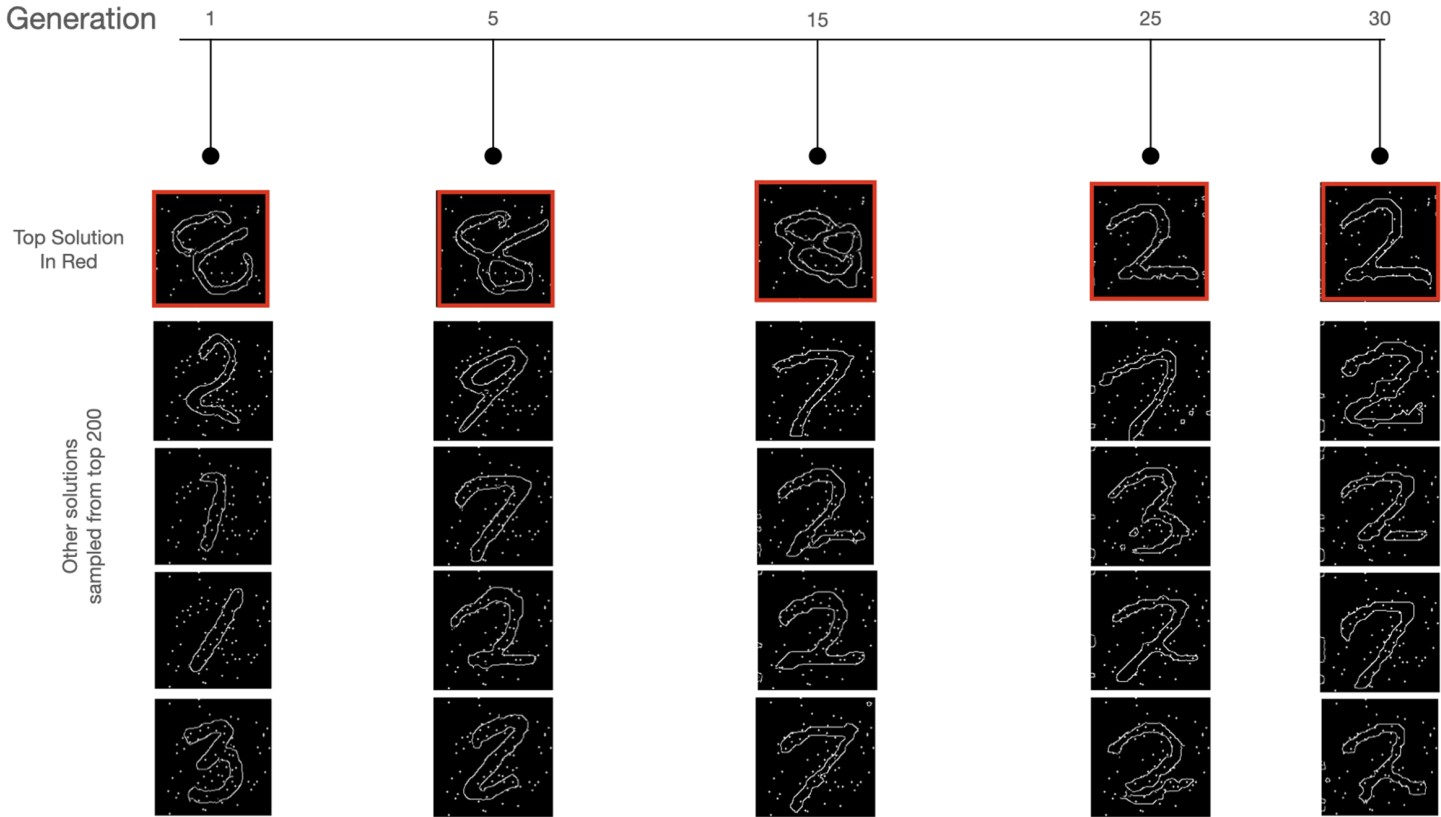

**Fig 6. Maintaining multiple hypotheses in the solution set.** In this example, we show how the correct solution (number 2) appears in the candidate pool of GenSearch before becoming the top solution. The correct solution appears already in generation 1 and with increased frequency in generations 5 and 15 before converging as the top solution in generation 25.

case a better shape fit is found using these numbers. We speculate that a similar process may occur in humans, as they also exhibited examples where multiple number candidates were tested (by drawing) before settling on one of their earlier guesses, indicating that multiple alternatives can be weighed in parallel.

## Comparison with gradient descent search

We implemented a version of GenSearch using gradient descent as the search algorithm instead of evolutionary search. This gives us a comparison to see the behavioural changes and results we obtain when only one hypothesis is greedily optimised by the search process. The results we obtained for both the MNIST and Fashion MNIST datasets for difficulty level 11 are given in Table 1. Fig 7 shows a target dot heatmap and the resulting images generated by the process. It also shows the optimisation of the solution during the gradient descent search.

In Table 1, we notice that the gradient-based search, on average, finds contours that pass through a larger number of dots compared to solutions by evolutionary search. However, its overall accuracy of finding correct shapes is significantly lower. Looking at the solutions of the gradient descent search in Fig 7 we see that it finds solutions with many artifacts that maximise the contour's fit to the dots. This leads us to hypothesise that one of the benefits of evolutionary search is that it regularises the search against overfitting to local artefacts. The fact that evolutionary search maintains multiple candidates along with the stochastic nature of operations on those candidates, i.e., selection, mutation and crossover, may help to produce this regularising effect against noisy and less stable shapes and artefacts.

**Table 1**. Accuracy and average dots covered for difficulty level 11 using gradient descent to find the optimal solution.

| Search Variant | MNIST | | Fashion MNIST | |
|---|---|---|---|---|
| | Accuracy | dots covered | Accuracy | dots covered |
| Genetic Search | 0.66 | 33 | 0.63 | 54 |
| Gradient Descent | 0.18 | 49 | 0.23 | 75 |

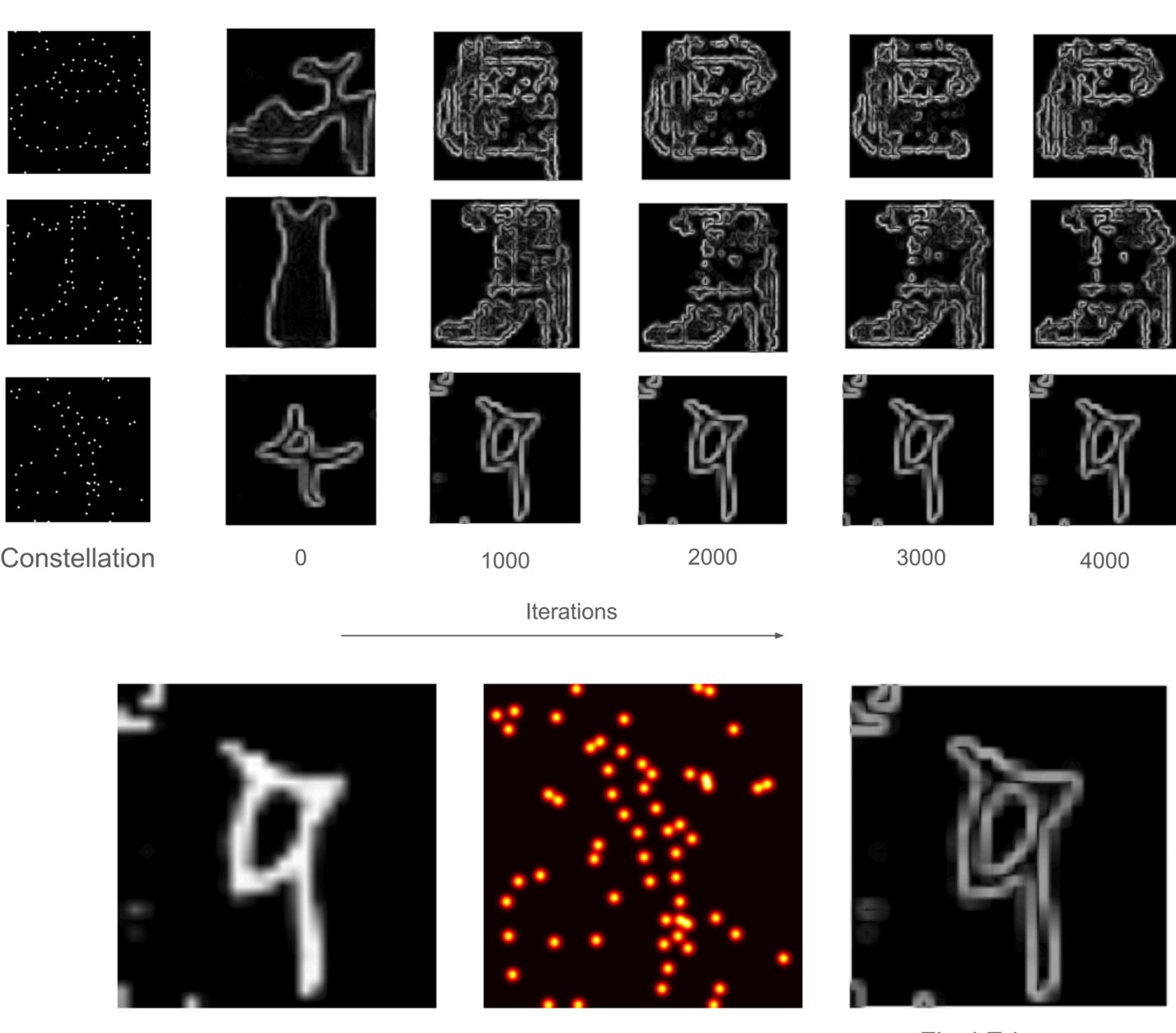

Constellation

0 1000 2000 3000 4000

Iterations

Final output Target Heatmap Final Edge

**Fig 7**. **Solving process of gradient-based search.** The upper panels show, for different constellation images, the change in candidate updated using gradient descent over the iterations. The lower panels show, using the final constellation image, that the objective of the search is to match the generated edge map, i.e., the lower panel rightmost image, to the headmap of dots, i.e., the middle image on the lower panel, at non-zero points of the heatmap.

## Comparison of solutions to machine learning models

We have seen the performance of our proposed GenSearch algorithm and how its performance and behaviour are similar to humans in the constellation task. However, we know that most of the top-performing models in practice are deep learning models. Also, such models based on CNNs have been proposed as prominent candidates for explaining aspects of human vision [13]. Hence, for comparison to GenSearch, we analyse the performance of machine learning models on the constellation task and evaluate how closely the mechanisms used by these models align with how humans solve the constellation images. In this section, we analyse CLIP [14] (Zero-shot classification), Resnet 18 (CNN trained on constellation images) and Pix2Pix [15] (Generative model trained on constellation images) and report how adequately these models capture human behaviour while solving the constellation images.

**Zero-shot performance using CLIP.** We first used zero-shot CLIP on the two constellation image sets at difficulty level 11 and found that the model's performance was near random (0.07 and 0.11) for MNIST and Fashion MNIST constellations. Given that CLIP was not explicitly trained for that modality, this outcome is expected. A detailed analysis of CLIP's performance across various type of images leading to constellation images can be found in [9].

**Trained Resnet 18 models on constellation datasets.** We further trained Resnet 18 models on both datasets for various difficulty levels. The training set in both cases comprises constellation images from 60,000 training images in the original dataset. For comparison across models, we use the same 38 test images for reporting test accuracy. We see in Fig 8a that it has a near-perfect accuracy that degraded with difficulty level but still is well above the accuracy shown by GenSearch and humans.

We attribute this high performance to training and thus overfitting for the particular modality. We further test the adaptability of the trained Resnet18 on difficulty levels other than its training set. We notice in Fig 8b that although the models adapt to images from difficulty levels close to their trained level, the performance drop is sharp in the difficulty level that differs from the original training. Notice that overfitting to a particular difficulty level is particularly evident when Resnet18 model trained on level 17 performs significantly worse on a much easier difficulty level of 9 or 11. On the other hand, humans and GenSearch show an expected performance trend, where their performance drops while going from easy to harder levels using the same algorithm.

We also observe that the models trained on challenging images, such as level 17, perform far above human accuracy, which is close to random at that difficulty level, indicative of a completely different statistical pattern of learning by these networks compared to the algorithm used by humans. Further, we analyse the Grad-Cam [16] based region attribution of the image to the Resnet18 model outputs (detailed result in supporting information in S3 Text), which shows that the model focuses on the presence of dots in some specific regions of the image for a particular class, which points to attributing spatial statistics of dots to a specific class rather than the structural analyses employed by humans and the GenSearch algorithm.

**Trained image-to-image model (pix2pix) on constellation datasets.** Pix2pix is an image-to-image model that learns to translate images from one modality to another. We use this method to solve the constellation images by learning to translate between a given constellation image and its corresponding object outline image. We train the model on the training set using the constellation images and their corresponding true object outline images for both the MNIST and Fashion MNIST datasets. The details of hyperparameters tuned to find the best model are given in S4 Text. The best models failed to learn a satisfactory mapping between the modalities, as seen in the best and worst solutions from the select test set in Fig 9. In Table 2, we also show the mean IOU(dots) of the solutions' overlap to the human drawings, and it achieves a low IOU(dots) of 0.25 (Fashion MNIST) and 0.23 (MNIST) compared to 0.49 and 0.6 for the GenSearch algorithm.

We further checked how a top-performing model trained on the MNIST dataset performs on Fashion MNIST constellation images and vice versa. We find that the models trained on a completely different dataset perform comparably to those trained on the specific dataset, i.e., the model trained on Fashion MNIST obtained an IOU (dots) score of 0.19 on MNIST constellations, and the MNIST-trained model had an IOU(dots) of 0.23 on Fashion MNIST. This indicates that

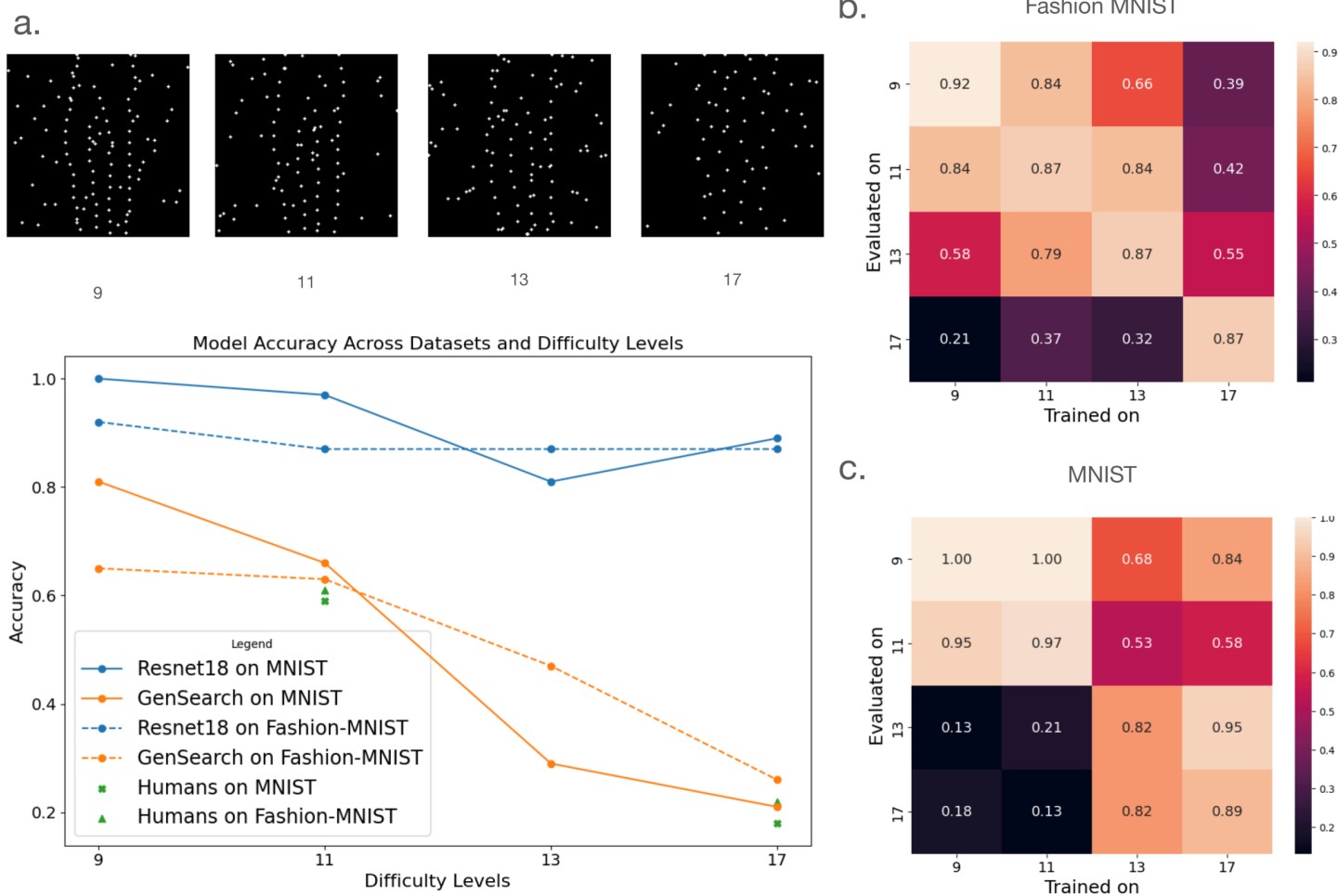

**Fig 8**. **Performance across difficulty levels.** a. The difficulty level of the constellation images is controlled by changing the distance between the dots, as shown in the example images in the figure. The plot shows the performance of the models and humans on different difficulty levels. Observe how the trained Resnet18 performs supernaturally compared to GenSearch and humans, especially on datasets with higher difficulty, i.e. level 17. b. Resnet 18 performance on change of train/test distribution for Fashion MNIST. Note how performance is unstable even on lower difficulty levels as it goes away from the training distribution. c. Resnet 18 performance on change of train/test distribution for MNIST

these models might be learning some local rules, e.g., connecting the dots at a certain distance, rather than relying on an understanding of the object categories depicted on the constellation images.

## Discussion

Visual categorization and recognition of objects can happen in a feed-forward manner. However, in complex visual tasks visual processing becomes more similar to the problem-solving process wherein top-down hypotheses are used to guide the search. Solving the constellation images with high noise usually involves steps akin to problem-solving, where various components must be analysed iteratively and in relation to each other to make sense of the scene [9]. Further, without traditional low-level cues for humans, the participants are forced to iteratively collect ambiguous low-level cues that fit the high-level description of the hypothesised object's shape.

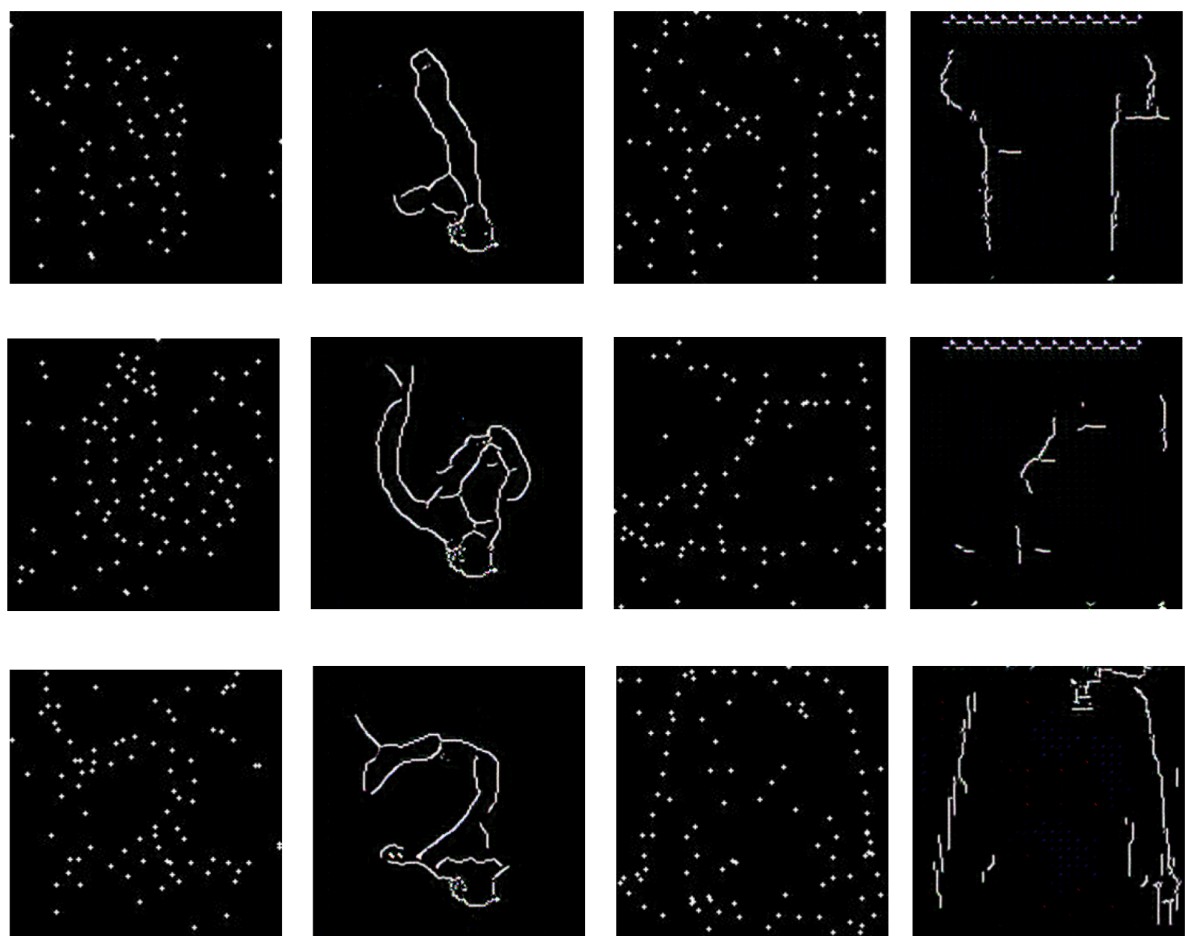

**Fig 9**. **Solutions by Pix2pix model.** The figure shows constellation images and their pix2pix solution on the right. The figures in the top row are representative of the best solutions by the model. IOU(dots) for both datasets is very low at 0.25 (Fashion MNIST) and 0.23(MNIST).

**Table 2**. **Average IOU (dots) and IOU (mistakes) between machine and human drawings.** IOU (dots) between human drawings and ground truth is 0.69 and 0.76 for MNIST and Fashion MNIST, respectively.

| Model | MNIST | | Fashion MNIST | |
|---|---|---|---|---|
| | IOU (dots) | IOU (mistakes) | IOU(dots) | IOU(mistakes) |
| GenSearch | 0.6 | 0.43 | 0.49 | 0.58 |
| Pix2Pix | 0.23 | 0.21 | 0.25 | 0.32 |

To represent this process computationally, we can consider this process as a search in the high-level object descriptions (Generator's latent space), iteratively refined by low-level cues (fitness of generated object's edges on the constellation image). In this work, the differential evolution process manages the prior update before generating a new set of hypotheses. We observe that this gives rise to similar solution dynamics as in humans, where particular solutions fitting a prominent feature of the object are maintained and further refined. GenSearch algorithm does not explicitly add additional weights (through the fitness function) to identified features that fit the dots and are likely to represent parts of objects. Still, implicitly, the top candidate selection, which is based on structural fit (i.e., fit to dots), encourages candidates that have parts fitting to dots to participate in the generation of the next population through point mutation (minor changes) and

cross-over (combination of two solutions). Currently, these mutations and crossovers do not explicitly preserve the partial fit. Still, the heuristics of differential evolution bias the search towards variations of partial fit solutions (as they are some of the top solutions of the previous generation) or to leave the partial fit only in pursuit of a better candidate. A large parent selection allows multiple candidate solutions to participate in the next generation, promoting diversity in the search space.

We also note some benefits of having the search as an evolutionary search mechanism, likely due to its ability to maintain multiple solutions during the search process. In our ablation of GenSearch's evolutionary search with gradient descent, we optimised for the same objective of passing the maximum number of dots. While both evolutionary search and gradient descent were able to find solutions that pass through almost an equal number of dots, evolutionary search found significantly more correct solutions, rather than overfitting to local artefacts. This experiment suggests that there is a regularising benefit of the multi-hypothesis stochastic nature of the evolutionary search.

The iterative refinement of visual solutions by humans has been highlighted in previous research [17–19] for visual problem-solving during ambiguous conditions. Computational studies have shown that artificial neural networks with recurrent connections better capture human cortical dynamics during vision than feedforward networks [20]. At the same time, previous experimental research also raises questions about managing multiple hypotheses concurrently in situations such as multi-stable visual perception [21]. As our results suggest, with the change in the top candidate at the end of each generation, usually, there are iterative changes to the same pattern that continue improving the solution. However, search heuristics also allow a completely new top solution to emerge, which is not iterative and disregards the top fit from the previous generation. A similar process in humans, where solutions can arise suddenly rather than through gradual refinement, often as a remote or unexpected idea takes precedence over a more conventional association, is well documented in the problem-solving literature as an "insight" [22–24]. Hence, aspects of GenSearch, such as maintaining multiple hypotheses, can provide a heuristic similar to that of humans considering different hypotheses in search of a visual solution. This mechanism also allows for multiple and even conflicting solutions to be preserved and take part in the solution process as long as they are not replaced by better candidates within allowed computational limits (controlled by the number of parents in the algorithm and population size in the algorithm). Overall, multi-hypothesis search algorithms like evolutionary search are a likely candidate to explain at least part of the search processes involved in human vision and reasoning.

GenSearch has several limitations and weaknesses. We observe that the mechanism heavily depends on the generator's ability to generate images at the right level of abstraction. While solving such images, humans reported identifying cues at various levels, such as 'lines or curves' or higher-level features like a 'wing' or 'handle', and then conditionally generating candidate shapes based on those initial features [9]. This points to a more flexible search process in humans than GenSearch, as humans can traverse between levels of feature hierarchies during the process whereas GenSearch only does a high-level search in GAN's latent space. As a result of this, with GenSearch, when objects are generated with too much detail, such as advertisement writing on the objects or patterns on the cloth or couch, the edge detector would consider everything as part of the candidate solution. And these spurious low-level features can misguide the search. Another limitation due to this lack of flexibility and guidance from different levels is the need for unnatural computational resources required by the search process, where it evaluates 1000 hypotheses per generation, leading to long inference times. These problems are especially exacerbated when using GenSearch on a more general constellations dataset with a diverse set of objects instead of MNIST and Fashion MNIST (discussed in supporting information in S1 Text). Such problems have been associated with the analysis-by-synthesis framework [25], and the proposed solution is a quick, low-level priors-based initialisation of search and fast refinement between high-level and low-level cues. For our case, technologies such as Control Net [26], which allow for generating images conditioned on specific parameters such as pose or outlines, can be explored as a relevant future direction for exploring the shape-conditioned generator to work more efficiently on large search spaces of natural images. On the other hand, our analysis of CNNs (specially trained Resnet18) and previous research [27] shows its over-reliance on learnt low-level local features that fit the training

modality. We infer that training CNN is like learning new bottom-up statistical cues that are highly effective for the current classification task but can lead to overfitting to the training distribution.

Our method's generalisation across difficulty levels matched better to humans, but GenSearch still does not match all nuances of human behaviour, such as reaction time. It also makes certain mistakes while solving that humans are unlikely to make (e.g. considering an unlikely object for the first iteration, as illustrated with the top candidate of trousers instead of a shoe in Fig 5). As the hyperparameters of the search algorithm were currently optimised to obtain the highest accuracy using the given algorithm, further research can optimise the algorithm for a better fit to the human process of solving. The variation in the breadth and depth of the search algorithm using the population size, along with the mutation and crossover types and rates, provide promising ways to explore different behaviours of this process. Furthermore, in our analysis, we compared our model against feedforward networks or gradient-based search. It will be useful for future studies to explicitly test the behaviour of recurrent neural networks, which have been shown to better capture human behaviour in terms of reaction times and performance trade-off [28] and outperform their feedforward counterparts under challenging conditions [29,30].

In conclusion, we developed a generative search algorithm that mimics key aspects of human visual processing while solving a complex vision task. The GenSearch algorithm provides a computational account of how candidate visual hypotheses might be generated, evaluated, and refined in tasks where bottom-up cues are insufficient. It is important to note that we are not claiming that the human brain necessarily implements genetic algorithms. Rather, the aim of this work was to explore whether a generative search approach can approximate how humans approach complex visual problem-solving. We show that GenSearch captures some dynamics that are observed in human behaviour, such as iterative hypothesis refinement, diversity of solutions, and occasional sudden shifts. These parallels suggest that search over generative representations may be a useful framework for understanding aspects of human visual inference in visually ambiguous contexts. Our research paves the way for further research for a more complete understanding of biological visual processing and how to mimic it in artificial systems.

## Materials and methods

### Ethics statement

The human experiments reported in this study were conducted in accordance with ethical research guidelines and received approval from the Research Ethics Committee of the University of Tartu with approval number 396/T-26. Participants were provided with relevant information verbally and in written form on the screen before beginning the task. To maintain anonymity, no written consent forms were collected, but all participants had to indicate their informed consent by pressing a button, before proceeding with the task.

### Experimental setup for generative search (GenSearch)

GenSearch consists of three major components: 1) image generator, 2) genetic search and 3) solution evaluator. The GenSearch algorithm uses dataset-specific components for its operation, such as an image generator and image classifier. These components are trained on the original images from the train set of the dataset. We evaluate GenSearch on the same 38 images from the MNIST constellations and Fashion-MNIST constellations dataset used in human experiments. All constellation images used in our experiments are single-channel with 160x160 resolution. During the evaluation, separate dataset-specific pre-trained components are used for each dataset, but the same search algorithm (including hyperparameters) is used. Implementation details of each of the components are given in the following section.

**Setup and training details for the generator.** The generator used for GenSearch is DCGAN [31] trained on the training set (60k images) of original MNIST and Fashion MNIST images. The two components of the model architectures are configured to have four deconvolution layers for the generator and four convolutional layers for the discriminator. The input noise size is 256, and the generated output image is configured for $64 \times 64$. Label smoothening is used during GAN

training. The generated images are scaled to $160 \times 160$ before the Canny edge detector is applied to obtain the candidate solution. The samples of images from the GANs are given in supplementary materials S5 Text.

**Genetic search implementation details.** The GenSearch algorithm is a genetic search on a generator's latent space implemented using PyGad [32] library. The search algorithms' hyperparameters and design decisions were tuned on a few samples from the train set images of datasets to optimise for best accuracy over images. The values of the tuned hyperparameters are given in Table 3. The tuned hyperparameters included the maximum number of generations, initial population size, mutation rate, crossover rate, convergence condition, and error tolerance between the dots and the line segment. We use the same hyperparameter setting for the model that works for both datasets. The fitness evaluation within a generation is parallelised with a batch size of 100 candidates. The inference algorithm takes close to 6 hours for the set of 38 test images. All experiments were run using Tesla V-100 GPUs. A setup with variation of these hyperparameters, testing GenSearch with lower population size in given in S6 Text.

**Setup for analysing solutions.** The search algorithm is guided by the fitness which is calculated by counting the number of dots the lines of a candidate solutions pass through. To analyse the drawing solutions, i.e., count the number of dots, we first obtain the position of dots in the constellation image. We then obtain the contours of the candidate solution outline using the findContours function from the OpenCV library. We count all the dots for the fitness with a distance of less than 3 pixels to a drawn contour. To obtain the category label for the candidate solution, we used the original generator output before getting the outline. A trained CNN classifier was used to classify the generated images into MNIST and Fashion MNIST categories. We used off-the-shelf classifiers from PyTorch's model hub for these datasets.

**Implementation details for GenSearch with gradient descent.** To create a version of GenSearch where one can obtain gradients with respect to the loss, i.e. the number of dots passing through the dots, we created an approximate loss that is differentiable and mimics this objective function. The main idea is to match the similarity of the candidate edge maps that are generated by the GAN to the constellation dots near the positions around the dots. We used a heatmap with a standard deviation of 3 around the dots in the original constellation image. The heatmap $H(x, y)$ generated from a set of dot coordinates $\{(x_i, y_i)\}_{i=1}^{N}$ is defined as:

$$H(x, y) = \sum_{i=1}^{N} \exp\left(-\frac{(x - x_i)^2 + (y - y_i)^2}{2\sigma^2}\right)$$

The edge generation is also made differentiable by convolving the candidate image generated by GAN with Sobel filter for edge detection. The edge map is masked to evaluate values only where the heatmap value $H(x, y) > 0$. The dot-product similarity loss between a target dot heatmap $D(x, y)$ and the generated masked edge map $E(x, y)$ is defined as:

$$\mathcal{L} = -\sum_{x,y} D(x, y) \cdot E(x, y)$$

Finally, we optimise this loss using gradient descent on the given loss using a triangular learning rate schedule.

**Table 3. Final hyper-parameters for genetic search.**

| Hyper-parameter | Value |
|---|---|
| Initial Population | 1000 |
| Number of Parents | 200 |
| Mutation Rate | 0.5 |
| Mutation Type | Random |
| Crossover Rate | 0.01 |
| Crossover Type | Uniform |
| Maximum Generations | 30 |

## Experimental setup for capturing human drawings

To obtain human data, we conducted separate experiments using two image datasets (MNIST and fashion MNIST). The design of the experiments was approved by the university ethics committee. All participants were adult volunteers who received no compensation for their involvement, and no personal data was collected. The first task included 10 participants, and 11 people participated in the second task, with no overlap between the two groups. Each participant viewed 38 images from the same category and was provided with a list of possible objects beforehand (numbers 0–9 for MNIST, or ten fashion-related objects for fashion MNIST). Their task was to identify the object in each image and draw the outline of the object. Additionally, participants rated certainty in their response, the difficulty of each image, and how suddenly the solution occurred, using a seven-point scale.

Following initial analyses, we asked six random participants from each experiment to solve the same set of images at a high difficulty level. This follow-up aimed to determine whether human solvers with some prior experience would be able to solve the images at a level comparable to the model's performance on images that are extremely difficult to solve for the untrained eye (see level 17 in Fig 8a). Since all the data was collected without personal identifiers, we did not link the responses from these follow-up measurements to the data from the first task.

## Deep learning models' training details

All deep neural network architectures (Resnet18 and pix2pix) evaluated in this work are trained on the training set of MNIST and Fashion MNIST constellations dataset. The test set numbers in the results section are on the selected set of 38 images for comparison to the human and GenSearch performance. The base Resnet 18 architecture is used, but the input layer is adapted to handle single channel constellations input. All models are trained for two epochs on the whole train set. A setup of training and testing regularised version of Resent 18 models is given in S7 Text. Similarly, for the Pix2pix model, we arranged the training and test sets as pairs of constellations and used the correct solution contour images. The model is trained with the base architecture for 10 epochs.

## Metrics

**IOU (dots).** We proposed a metric specifically useful for the constellations dataset to check the alignment between different solutions. This metric is similar to the popularly used IOU metric in image segmentation, which measures the overlap between the generated and true segmentation masks. In our case, we measure a similar overlap in terms of the dots passed by the solutions A and B on the constellations image. The number of dots that the contour in drawing A passes through (with a margin of 3 pixels for all solutions) is n(A), and the number of dots passing solution B is n(B). The number of dots part of both solutions is $n(A \cap B)$. The Intersection over union is defined as

$$IOU(dots) = \frac{n(A \cap B)}{n(A \cup B)}$$

which can be written as

$$IOU(dots) = \frac{n(A \cap B)}{n(A) + n(B) - n(A \cap B)}$$

## Supporting information

**S1 Text. Adaptation of GenSearch on more complex THINGS dataset.**
(PDF)

**S2 Text. GenSearch's convergence vs human reaction time.**
(PDF)

**S3 Text. Additional analysis of Resnet18 models.**
(PDF)

**S4 Text. Hyperparameter tuning for pix2pix models.**
(PDF)

**S5 Text. Examples of images generated by GANs.**
(PDF)

**S6 Text. Evaluation of GenSearch with lower population size.**
(PDF)

**S7 Text. Evaluation of regularised Resnet 18 models that match human accuracy.**
(PDF)

## Acknowledgments

We thank Raul Vicente and Meelis Kull for the valuable discussions and inputs.

## Author contributions

**Conceptualization:** Tarun Khajuria, Jaan Aru.

**Data curation:** Tarun Khajuria, Kadi Tulver.

**Formal analysis:** Tarun Khajuria.

**Funding acquisition:** Jaan Aru.

**Investigation:** Tarun Khajuria, Kadi Tulver.

**Methodology:** Tarun Khajuria, Kadi Tulver.

**Software:** Tarun Khajuria, Kadi Tulver.

**Supervision:** Jaan Aru.

**Validation:** Tarun Khajuria.

**Visualization:** Tarun Khajuria.

**Writing – original draft:** Tarun Khajuria, Kadi Tulver, Jaan Aru.

**Writing – review & editing:** Tarun Khajuria, Kadi Tulver, Jaan Aru.

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
