## [Decision Letter · Decision Letter 0]

4 Jun 2025

PCOMPBIOL-D-25-00510

Comparing a computational model of visual problem solving with human vision on a difficult vision task.

PLOS Computational Biology

Dear Dr. Khajuria,

Thank you for submitting your manuscript to PLOS Computational Biology. After careful consideration, we feel that it has merit but does not fully meet PLOS Computational Biology's publication criteria as it currently stands. Therefore, we invite you to submit a revised version of the manuscript that addresses the points raised during the review process.

Please submit your revised manuscript within 60 days Aug 04 2025 11:59PM. If you will need more time than this to complete your revisions, please reply to this message or contact the journal office at ploscompbiol@plos.org. Please include the following items when submitting your revised manuscript:

We look forward to receiving your revised manuscript.

Kind regards,

Tim Christian Kietzmann, Dr. rer. nat.

Academic Editor

PLOS Computational Biology

Hugues Berry

Section Editor

PLOS Computational Biology

**Journal Requirements:**

At this stage, the following Authors/Authors require contributions: Tarun Khajuria, Kadi Tulver, and Jaan Aru. Please ensure that the full contributions of each author are acknowledged in the "Add/Edit/Remove Authors" section of our submission form.

4) Please provide a detailed Financial Disclosure statement. This is published with the article. It must therefore be completed in full sentences and contain the exact wording you wish to be published.

1) Please clarify all sources of financial support for your study. List the grants, grant numbers, and organizations that funded your study, including funding received from your institution. Please note that suppliers of material support, including research materials, should be recognized in the Acknowledgements section rather than in the Financial Disclosure

2) State the initials, alongside each funding source, of each author to receive each grant. For example: "This work was supported by the National Institutes of Health (####### to AM; ###### to CJ) and the National Science Foundation (###### to AM)."

3) State what role the funders took in the study. If the funders had no role in your study, please state: "The funders had no role in study design, data collection and analysis, decision to publish, or preparation of the manuscript."

4) If any authors received a salary from any of your funders, please state which authors and which funders..

5) Please send a completed 'Competing Interests' statement, including any COIs declared by your co-authors. If you have no competing interests to declare, please state "The authors have declared that no competing interests exist". Otherwise please declare all competing interests beginning with the statement "I have read the journal's policy and the authors of this manuscript have the following competing interests"

**Reviewers' comments:**

Reviewer's Responses to Questions

**Comments to the Authors:**

Reviewer #1: I found the motivation of this line of research difficult to follow. The task is so difficult it is difficult to know how it relates to the challenges the human visual system confronts. The authors attempt to link the task to more familiar problems such as when the write:

“This approach allows us to use static images for a dynamic and complex image processing task involving iterative steps, tackling a computational problem similar to visual inference and control in a dynamic setting such as for self-driving cars, where each inference step depends on the prior context.”.

But it is hard to see the relation of this task to the process of visual processing and processing of dynamic images. If humans could easily identify these images, then the logic of studying these images would make more sense to me – then DNNs would also need to support these visual functions if they are to be models of human vision. But the importance of matching quite poor performance is less clear to me. The authors need to do a better job motivating the phenomenon they are studying.

It is also the case the proposed solution of the model seems entirely unrelated two what how humans like reason under these conditions (trying to find lines in the noise). The author write:

“The constellations image is solved by generating candidate solutions with a GAN and refined using a genetic search conditioned on best fitting of the solution outlines to the dots on the constellation image”.

Is the claim that that humans also use GANs and genetic search? If not, what is the claim? The analyses of the accuracy levels, errors and confusion, does not seem (at least to me), sufficient to lend any support to the claim that human visual system is solving the tasks using similar algorithms.

The authors also claim that their model does a better job than a ResNet18 model given that performance was too high when trained on these images (or too low for Clip on 0-shot testing). But ResNet18 could have been given less training to get training set to human-like levels. Would error patterns be similar under these conditions? In any case, not suggesting this be done, but not sure the error patterns the author relying on provide strong support for their algorithm.

Reviewer #2: I did enjoy reading the manuscript. I think it can be improved in many ways -

1. The description of the algorithm in lines 85-103 is super useful but needs more clarification.

a. "The initial population is randomly sampled..". Population of what exactly?

b. Needs a step that generates images with the GAN

c. Steps 2 and 3 should perhaps be combined

d. Confusing usage of "point mutation and crossover" without explaining what it means first.

2. Can the correlation between confusion matrices be calculated for humans and resnet18? Reaction time could also be predicted with a recurrent neural network (Goetschalckx et al. 2023)

Reviewer #3: Khajuria, Tulver and Aru propose a way to formalize the idea of visual perception as an “analysis-by-synthesis” process, in which a generative model is used to propose solutions to a perceptual problem, which are then refined based on their compatibility with visual input. Here, this process is implemented as an evolutionary search over latent vectors in a pre-trained generative model (a GAN). The task is to interpret “constellation” images, in which an object’s outline is shown as a collection of dots amidst randomly positioned nuisance dots, as objects belonging to one of a set of categories. Specifically, images from the popular computer vision datasets MNIST and FashionMNIST are used to generate the stimuli. Vectors in the GAN’s latent space are evaluated by their compatibility with the dot arrangement shown in a given image. This provides a fitness function by which the most promising vectors are selected, which then undergo crossover and mutation. This process is iterated for 30 generations, yielding vectors that are well-adapted to the image. Classifying the underlying object category from the final solution images yields an accuracy that is similar to that of human subjects (~60% for both MNIST and FashionMNIST). The classification errors made by this model are also similar to those of humans. Moreover, the object shapes inferred by the model were also similar to those drawn by human subjects. The model is compared with a set of baseline deep learning models, finding that these models either failed at the task, or exhibited unrealistically high accuracy.

I found the approach described in this paper quite original and interesting, but I also feel that there are a few issues that the authors should address.

My main question is a conceptual one: what are the claims precisely, with respect to the search process? The authors have chosen to implement the search over latent parameters as an evolutionary algorithm. This is a very interesting choice, as search algorithms such as evolutionary search are more commonly used for discrete tasks with a combinatorial structure, such as planning or program synthesis. This choice has several implications. Particularly, the main implication is the ability to maintain a “library” of diverse solutions to the search problem. This is in contrast, for example, to gradient descent, which would more greedily explore a specific direction in the search space. The authors hint at the possibility that a more open-ended evolutionary search might be a better model of perceptual analysis-by-synthesis in humans, for example by mentioning multi-stable phenomena in vision, but this claim is not made explicit nor explicitly tested. In fact, I believe that the specific choice of an evolutionary algorithm is the main novelty of this work, as the idea of analysis-by-synthesis per se has been explored in depth already, but to my knowledge never implemented in this particular way. So I think an explicit comparison with a different search method would substantially strengthen the paper. The main alternative hypothesis, in my opinion, would be a search for latent parameter configurations via gradient descent, for two reasons: 1) Test-time optimization via gradient descent has been extensively used in energy-based models, showing that it is a viable method to solve many perceptual and reasoning tasks (Mordatch 2018, for example, used gradient descent to arrange dots into configurations that matched a particular goal concept, which might bear some resemblance to the task of finding objects in noisy dot configurations used here). 2) While there are indeed, as the authors point out, phenomena in human perception that suggest multiple explanations of a perceptually ambiguous stimulus are being considered, such as multi-stable stimuli, most computational models of perceptual ambiguity resolution (including object recognition in clutter) through recurrence in neural networks formalize the task as one of evidence accumulation in favor of a specific hypothesis (e.g. Thorat, Aldegheri & Kietzmann 2021, Goetschalckx, Govindajaran et al. 2023). While subtle differences in task design might lead to different search algorithms being more effective, this shows that the alternative hypothesis of a greedy search in one promising direction, whether implemented through gradient descent or recurrent dynamics, is worth testing. The comparison between evolutionary and gradient descent-based search could be implemented by making the pipeline from latent noise vector to contours fully differentiable. The GAN’s generator is already a neural network, and the edge detection could be replaced by a convolutional layer with fixed Gabor-like kernels. Both of these could be frozen, and gradient descent on the latent vector itself could be used to find optimal solutions. Of course, the authors should let me know in case there are practical difficulties with this approach which I am missing.

A more methodological issue regards the analysis measuring the overlap between GenSearch and human drawings. It is hard to make sense of an absolute IOU quantity, without a baseline to compare it with. The most straightforward baseline would be to add the IOU between human drawings and the ground-truth contours, to show whether GenSearch overlaps with human drawings merely because they are both trying to get at the same answer (IOU(humans, groundtruth) ≥ IOU(humans, gensearch)), or if it also makes similar mistakes (IOU(humans, groundtruth) < IOU(humans, gensearch)). Additionally, to clarify the same question, the correlation between IOU on individual images for GenSearch and humans should be reported, to show whether humans and GenSearch tend to perform worse or better on the same images (or categories, but individual images might be a more fine-grained and thus sensitive analysis).

I find the comparison with pix2pix particularly interesting and informative, since it looks like a perfectly suited model for the task which is based on completely different principles from the GenSearch model. However, it is hard to interpret its low performance. Are the authors confident that it is due to an intrinsic incompatibility between the model and this task, and not e.g. to a poor choice of hyperparameters? Within the limitations of the computational power at their disposal, the authors should try to explore hyperparameters to make pix2pix as competitive as possible: right now, its performance is very poor, as shown in Figure 8, making it hard to meaningfully compare it with humans. Of course, it is possible that the model is intrinsically unsuited for this task, so I’m not requiring the authors to reach a given level of accuracy. But at least, showing that its poor performance is consistent across hyperparameters would make the claims about the superiority of GenSearch in approximating human behavior more convincing.

Beyond these major points, below I list some minor and more specific issues that the authors should address:

- In Figure 2b, the bar plot does not include error bars. Adding them would be good scientific practice, at least for human subjects data.

- In Figure 4, which shows human mistakes next to GenSearch mistakes, it would be useful to see the ground-truth contours as well, to make it clearer what exactly the mistake was. Also, for this and other figures, including a few more examples would be good (unless the journal formatting guidelines make that difficult).

- In general, I find the human experiment sample sizes to be on the low side, but in particular, I do not think that having only two participants perform the task at the difficult level is sufficient. Unless there are major practical impediments in doing so, the authors should collect a larger sample on this difficulty level, and if possible on other difficulty levels (e.g. 9, 13) such that the results in Figure 7a can be more properly assessed, with confidence intervals as well.

- Somewhere either in the main text or supplementary materials, examples of the full color images generated by the GAN should be added, to get a clearer idea of the workings of the model. Following the analogy implied by this model, the GAN outputs would be something like the mental images of the object evoked by the constellation stimuli, so seeing these images would be an interesting qualitative measure of how humanlike the model is.

- At line 357, p. 15, there is a typo: the table number is missing at the very end of the line.

References:

- Goetschalckx, L., Govindarajan, L. N., Karkada Ashok, A., Ahuja, A., Sheinberg, D., & Serre, T. (2023). Computing a human-like reaction time metric from stable recurrent vision models. Advances in neural information processing systems, 36, 14338-14365.

- Mordatch, I. (2018). Concept learning with energy-based models. arXiv preprint arXiv:1811.02486.

- Thorat, S., Aldegheri, G., & Kietzmann, T. C. (2021). Category-orthogonal object features guide information processing in recurrent neural networks trained for object categorization. arXiv preprint arXiv:2111.07898.

**Have the authors made all data and (if applicable) computational code underlying the findings in their manuscript fully available?**

Reviewer #1: None

Reviewer #2: None

Reviewer #3: Yes

PLOS authors have the option to publish the peer review history of their article (what does this mean?). If published, this will include your full peer review and any attached files.

Reviewer #1: No

Reviewer #2: No

Reviewer #3: **Yes: **Giacomo Aldegheri

**Figure resubmission:**
---

## [Decision Letter · Decision Letter 1]

12 Sep 2025

PCOMPBIOL-D-25-00510R1

Comparing a computational model of visual problem solving with human vision on a difficult vision task.

PLOS Computational Biology

Dear Dr. Khajuria,

Thank you for submitting your manuscript to PLOS Computational Biology. After careful consideration, we feel that it has merit but does not fully meet PLOS Computational Biology's publication criteria as it currently stands. Therefore, we invite you to submit a revised version of the manuscript that addresses the points raised during the review process.

Reviewers#1 and #3 are still not entirely satisfied with your answers to the points they raised. We agree that the points raised by reviewer#3 are indeed significant and should be accounted for in a revised version. The points of rev#1 appear also very good, in particular about the comparison to a non-overfitted ResNet18. Comparison to the performance of other network architectures is certainly a very good suggestion too but could probably be adequately addressed as a perspective/discussion in the text.

Please submit your revised manuscript within 60 days Nov 12 2025 11:59PM. If you will need more time than this to complete your revisions, please reply to this message or contact the journal office at ploscompbiol@plos.org. Please include the following items when submitting your revised manuscript:

We look forward to receiving your revised manuscript.

Kind regards,

Tim Christian Kietzmann, Dr. rer. nat.

Academic Editor

PLOS Computational Biology

Hugues Berry

Section Editor

PLOS Computational Biology

**Journal Requirements:**

Please ensure that the CRediT author contributions listed for every co-author are completed accurately and in full.

At this stage, the following Authors/Authors require contributions: Tarun Khajuria, Kadi Tulver, and Jaan Aru. Please ensure that the full contributions of each author are acknowledged in the "Add/Edit/Remove Authors" section of our submission form.

**Reviewers' comments:**

Reviewer's Responses to Questions

**Comments to the Authors:**

Reviewer #1: I think the authors for responding to my questions, I understand the motivation of the project better now. But I’m still not convinced that that the findings merit publishing in PLOS Computational Biology. I’m not confident in my assessment, but let me outline my outstanding concerns.

The authors have clarified how the constellation task relates to visual problem solving that the visual system confronts – it is thought analogous to “trying to identify the approaching person in a dark alley... as it requires combining low-quality bottom-up cues with prior knowledge and iterating over many possibilities”

This makes more sense to me now, but I’m not convinced the authors have provided a strong enough test of the correspondence of their algorithm. The model can match overall performance, shows a similar reduction in performance with increased difficultly, and shows a similar pattern of errors (such that similar shapes can be confused). This is contrasted with a ResNet18 that performs near ceiling across levels of noise, which is taken to reflect overfitting. 

But presumably it would be quite easy to reduce overall performance of the ResNet18 models, such as providing it less training, and attempt to reduce overfitting by adding some regularisation terms, or dropout, for example. Whether the advantages would still be obtained when more of an effort was testing alternative approaches is not clear to me.

The authors also clarify their theoretical claim, writing: “It is important to note that we are not claiming that the human brain necessarily implements genetic algorithms, rather the aim of this work is to explore whether a generative search approach can approximate how humans approach complex visual problem-solving.”.

So the claim is much more general than this particular algorithm they report, or even evolutionary approach, and I wonder whether this suggests that other architectures would succeed, such as recurrent networks. Indeed, it has been shown that recurrent networks are better able to solve more difficult visual problems, and that these models can converge to solutions rather than a fast single bottom-up pass (Nayebi et al., 2022). Clearly, humans are doing in problem solving context the authors are studying – making multiple attempts to solve the task. So, it feels like the authors are comparing their approach to a bit of a strawman model (ResNet), and that other theoretically motivated approaches might succeed to the same extent.

Nayebi, A., Sagastuy-Brena, J., Bear, D. M., Kar, K., Kubilius, J., Ganguli, S., ... & Yamins, D. L. (2022). Recurrent connections in the primate ventral visual stream mediate a trade-off between task performance and network size during core object recognition. Neural Computation, 34(8), 1652-1675.

Reviewer #2: Thank you for addressing my comments.

Reviewer #3: I thank the authors for taking the time to address my concerns thoroughly. I do think the paper has been substantially improved. I still have concerns, however, about the way the (updated) results are presented. I think in its current form the manuscript lacks clarity. My concerns regard two results in particular: the comparison with gradient descent, and the IOU results.

- Comparison with gradient descent: the authors have done an excellent job in making the pipeline differentiable and comparing the evolutionary search results with those obtained using gradient descent. I think, however, that these results should be made clearer in the following ways:

1. As far as I can tell, right now only a single example of the gradient-based search results is shown in Fig. 7. Including a few more examples would help the reader to gauge systematic differences between this and the evolutionary search.

2. In Table 1, only the classification accuracy for the evolutionary vs. gradient-based version of GenSearch are reported. In the text, the authors also mention that the gradient-based search obtained a higher overlap score with the dots, but this score is not reported. It should be added to the table, next to the classification accuracy, to highlight the tradeoff.

3. In reporting the results of the gradient-based model, the authors write (p. 10, lines 210-212): “We observe that the gradient-based search finds solutions optimal to the desired objective, i.e., the search increases the number of dots passing through the candidate solution least as high as the top solution obtained using evolutionary search.” This sentence is not grammatical, so I would invite the authors to rephrase it in a clearer way.

4. The authors make claims about the ability of evolutionary search to maintain multiple solutions being key to its success in approximating human performance. They also report the best-performing hyper-parameters of the genetic algorithm in Table 3. It might be useful to clarify to what extent these hyper-parameters are consistent with the hypothesis that the ability to maintain multiple hypotheses is central to the success of GenSearch (e.g. larger mutation rate, larger solution pool or population size…). On p. 15, lines 314-315, the authors write “A large parent selection allows multiple candidate solutions to participate in the next generation, promoting diversity in the search space”, which gets close to the idea that this selection of hyper-parameters works well because of its ability to maintain diversity, but this is not tested explicitly.

5. In general, I feel like there are many claims about similarities between GenSearch and human cognition which are not substantiated by the data. One example is the aforementioned ability to maintain multiple solutions: the only evidence for this being related to the search process in humans are the two examples in Figure 5. While these look compelling, two individual examples are not sufficient to draw broad conclusions. Moreover, other tests of this similarity in search dynamics, namely the correlation between convergence and reaction times, yield inconclusive results. I would thus encourage the authors to either (A) conduct more thorough tests of the relation between GenSearch and the search for the best-fitting solutions by human subjects, or (B) express any claims about similarities, or mechanistic explanations for GenSearch’s superior performance to other methods, in more tentative terms. For example, on page 15, lines 322-323, they write that GenSearch’s superior classification performance to gradient descent “showed the regularising benefit of the multi-hypothesis stochastic nature of the evolutionary search”. As there are several differences between evolutionary search and gradient descent beyond the ability to hold multiple hypotheses, I would rephrase this as “suggests” rather than “showed”. Similarly, on p. 16, lines 337-339 “GenSearch, by maintaining multiple hypotheses, provides a natural heuristic similar to that of humans considering different hypotheses in search of a visual solution”. While I think it is fine to speculate that considering different hypotheses is what makes GenSearch’s performance superior to gradient descent (in fact I suggested using gradient descent as a baseline for this reason!), I don’t think the present data justify such strong claims.

- IOU results: in their response, the authors do mention the IOU between ground-truth and human drawings. I believe that this would be informative to include in the table (or turn the table into a figure), as an upper bound. Also, it is informative to show that the overlap between human drawings and ground-truth is still higher, showing that humans are still better than the model at capturing the shape of the ‘hidden’ objects. Perhaps, but this is just a suggestion, it could even be interesting to show this at different levels of difficulty, since with more noisy and ambiguous stimuli human subjects are likely to rely on their priors more, and GenSearch might be able to capture these priors. Concerning the overlap of mistakes, I think this should also be reported in the figure/table together with the other IOU results, to get a clear and complete pictures. Even more importantly, the overlap in mistakes between pix2pix and humans should also be included, to get an apples-to-apples comparison with GenSearch.

In general, I stand by my initial assessment that this is a valid and interesting work. I do think, however, that it is more of a proof-of-concept than a conclusive finding. With the modifications I recommended, including a change in wording to highlight the speculative nature of some conclusions, I believe it will be ready for publication.

**Have the authors made all data and (if applicable) computational code underlying the findings in their manuscript fully available?**

Reviewer #1: Yes

Reviewer #2: Yes

Reviewer #3: Yes

PLOS authors have the option to publish the peer review history of their article (what does this mean?). If published, this will include your full peer review and any attached files.

Reviewer #1: No

Reviewer #2: No

Reviewer #3: **Yes: **Giacomo Aldegheri

**Figure resubmission:**
---

## [Decision Letter · Decision Letter 2]

21 Nov 2025

Dear Mr Khajuria,

We are pleased to inform you that your manuscript 'Comparing a computational model of visual problem solving with human vision on a difficult vision task.' has been provisionally accepted for publication in PLOS Computational Biology.

Best regards,

Tim Christian Kietzmann, Dr. rer. nat.

Academic Editor

PLOS Computational Biology

Hugues Berry

Section Editor

PLOS Computational Biology

Reviewer's Responses to Questions

**Comments to the Authors:**

Reviewer #1: I think the revisions have improved the paper, but I continue to have my doubts that the manuscript merits publication here. I don’t think the following response to one of my concerns is adequate. That said, I’m not confident in my assessment, and happy to defer to others.

Jeff Bowers

My previous comment:

So the claim is much more general than this particular algorithm they report, or even evolutionary

approach, and I wonder whether this suggests that other architectures would succeed, such as recurrent

networks. Indeed, it has been shown that recurrent networks are better able to solve more difficult visual

problems, and that these models can converge to solutions rather than a fast single bottom-up pass

(Nayebi et al., 2022). Clearly, humans are doing in problem solving context the authors are studying –

making multiple attempts to solve the task. So, it feels like the authors are comparing their approach to a

bit of a strawman model (ResNet), and that other theoretically motivated approaches might succeed to

the same extent.

Response: Thank you for raising this issue. We have now added this clarification to the

discussion as a limitation and direction for future work: Furthermore, in our analysis, we

compared our model against either feedforward networks or gradient-based search. It will be

useful for future studies to explicitly test the behaviour of recurrent neural networks, which have

been shown to better capture human behaviour in terms of reaction times and performance

trade-off (Spoerer et al., 2020) and outperform their feedforward counterparts under challenging

conditions (Spoerer et al., 2017).

Reviewer #2: I think it is an interesting implementation of a analysis-by-synthesis model and I support its acceptance.

Reviewer #3: Thank you for addressing my remaining concerns. I believe the work has been substantially strengthened, and it is now ready for publication.

**Have the authors made all data and (if applicable) computational code underlying the findings in their manuscript fully available?**

Reviewer #1: Yes

Reviewer #2: None

Reviewer #3: Yes

PLOS authors have the option to publish the peer review history of their article (what does this mean?). If published, this will include your full peer review and any attached files.

Reviewer #1: **Yes: **Jeffrey Bowers

Reviewer #2: No

Reviewer #3: **Yes: **Giacomo Aldegheri

---

## [Editor Report · Acceptance letter]

PCOMPBIOL-D-25-00510R2

Comparing a computational model of visual problem solving with human vision on a difficult vision task.

Dear Dr Khajuria,

I am pleased to inform you that your manuscript has been formally accepted for publication in PLOS Computational Biology. Your manuscript is now with our production department and you will be notified of the publication date in due course.

With kind regards,

Judit Kozma
